# Examining Loneliness: A Comparative Analysis of Face-to-Face, Telephone, and Online Communication among Japanese Young Adults

Yuko Tanaka [1,*], Yuka Iwata [2], Nanami Oe [3] and Etsuko Tadaka [3,*]

1 Department of Community Health Nursing, School of Nursing and Social Services, Health Sciences, University of Hokkaido, Tobetsu-cho, Ishikari-gun 061-0293, Japan
2 Department of Community Health Nursing, Graduate School of Medicine, Yokohama City University, Yokohama 236-0004, Japan; iwata.yuk.go@yokohama-cu.ac.jp
3 Department of Community and Public Health Nursing, Graduate School of Health Sciences, Hokkaido University, Sapporo 060-0812, Japan; o_nanami0706@pop.med.hokudai.ac.jp
* Correspondence: yta@hoku-iryo-u.ac.jp (Y.T.); e_tadaka@pop.med.hokudai.ac.jp (E.T.)

**Abstract:** (1) Background: Loneliness is inherently linked to social connections, with interpersonal communication playing a pivotal role. Despite this connection, limited research exists on the relationship between loneliness and communication among young adults. This study investigates the correlation between face-to-face, telephone, and online communication frequencies and loneliness among individuals in their 20s. (2) Methods: Using a dataset from a nationwide survey conducted by the Japanese Cabinet Office, this study focuses on 1812 respondents aged 20–30, selected from a random sample of 20,000 individuals aged 16 and older across Japan. (3) Results: A Multivariate Logistic Regression Analysis reveals a significant association between communication frequency and loneliness, even after adjusting for demographic characteristics. Notably, decreased communication frequency across all modalities correlated with increased loneliness. Online communication exhibited the highest impact, followed by face-to-face communication, with phone call communication ranking last. (4) Conclusions: This study emphasizes the importance of seamlessly integrating social networking service (SNS)-based communication with various forms of social interaction. A well-balanced integration of these approaches is crucial for mitigating loneliness among young individuals and promoting positive mental health outcomes.

**Keywords:** loneliness; young adult; communication; online; face-to-face; telephone

## 1. Introduction

Loneliness is "an unpleasant experience that occurs when an individual's network of social relationships is deficient in some important way, either quantitatively or qualitatively" (Peplau and Perlman 1982). This underscores the critical need to develop comprehensive strategies to address these issues across societal domains. Recognizing this imperative, Japan took a significant step by enacting "The Act Promotion of Policy for Loneliness and Isolation" (Japan 2023), making it the second advanced nation after the United Kingdom (United Kingdom 2018) to implement such legislation. This Act seeks to create a society in which not a single person suffering from loneliness and isolation is left behind. Furthermore, it aims to foster an environment in which individuals actively support and connect, catalyzing the inception of a nationwide survey aimed at scrutinizing the landscape of loneliness and social isolation.

This groundbreaking survey, encompassing a random selection of 20,000 individuals aged 16 and older, garnered responses from 11,218 participants, constituting 56.1% of the surveyed population. The results, assessed through the UCLA Loneliness Scale (Arimoto and Tadaka 2019), revealed a nuanced understanding of loneliness prevalence

across different age groups. Notably, individuals in their 20s and 30s exhibit a significantly higher prevalence of loneliness. In the 20–29 age group, 9.4% reported constant loneliness, and 41.9% experienced intermittent loneliness. Similarly, in the 30–39 age group, 10.0% reported constant loneliness, with intermittent loneliness at 43.7%. These findings underscore age-related differences in loneliness, emphasizing a distinct pattern of heightened prevalence among individuals in their 20s and 30s (Cabinet Secretariat Japan 2021–2022). The phenomenon of higher loneliness among younger individuals than older adults is consistent with large-scale surveys conducted by the BBC (Barreto et al. 2021) and the Community Life Survey in the UK (England, Office for National Statistics 2018).

The 20s and 30s age groups exhibited disparities in marriage, employment, economy, and health status. Individuals in their 20s represent a crucial period marked by physical, psychological, and social development, where career formation, new relationships, and social connections evolve. Notably, this phase often leads to heightened susceptibility to feelings of loneliness (Lim et al. 2020). A qualitative study targeting young individuals in London reported that loneliness can serve as a positive experience by providing time and space for introspection and opportunities for growth and development. However, if experienced for an extended period, loneliness has the potential to be harmful (Fardghassemi and Joffe 2021). The persistence of loneliness adversely affects the health and well-being of individuals in their 20s and has the potential for long-term consequences in the future. Given that individuals in their 20s play a pivotal role as future contributors to society, addressing and preventing loneliness in this demographic is paramount to the overall prosperity of communities.

Loneliness is associated with the onset of and mortality from cardiovascular diseases, brain disorders, and other chronic illnesses (Friedler et al. 2015; Valtorta et al. 2016). In the younger generation, it has been linked to decreased sleep quality (Cacioppo et al. 2002), development of chronic illnesses (Christiansen et al. 2021), and the occurrence of depressive symptoms and mental disorders (Christiansen et al. 2021; Vanhalst et al. 2012), indicating connections to both physical and mental health issues.

Regarding gender differences in loneliness, reports suggest that young adult males experience significantly higher levels of loneliness than females (Barreto et al. 2021). However, research through meta-analyses does not demonstrate clear gender differences throughout life (Maes et al. 2019).

The relationship between loneliness and household composition reveals that living alone is associated with higher levels of loneliness among young adults (Franssen et al. 2020). Regardless of the presence of cohabitants, limited support from romantic partners or friends (Lee et al. 2018), and a limited social circle or low social involvement are linked to increased feelings of loneliness (Luhmann and Hawkley 2016). This suggests the significance of social connections among young adults, in conjunction with their household situation.

Regarding the association between loneliness and employment status, employed young individuals tend to experience relatively lower levels of loneliness than their unemployed peers (Luhmann and Hawkley 2016). The perception of a financial imbalance contributes to heightened loneliness (Franssen et al. 2020). Stable employment contributes to socioeconomic stability and mitigates loneliness.

Loneliness arises from the discrepancy between actual and desired social relationships (Peplau and Perlman 1982), necessitating consideration from a sociocultural perspective. A cross-cultural survey conducted in five European democracies found that higher perceptions of collectivism were associated with decreased loneliness (Heu et al. 2019), suggesting collectivism as a potential buffer against loneliness. However, research indicates that neoliberalism, by diminishing the sense of connection and emphasizing competition, leads to increased loneliness, resulting in decreased happiness (Christiansen et al. 2021). While individualism has become more prevalent in Japanese culture over time, an acknowledgment that collectivist values persist exists (Ogihara 2017). A study examining cultural differences in seeking social support (Zheng et al. 2021) revealed that Japanese individuals were less inclined to seek social support during stressful periods than European Americans, indicat-

ing higher levels of loneliness (Zheng et al. 2021). The cultural background of the Japanese population being less inclined to seek social support may potentially influence a heightened sense of loneliness. Exploring the experience of loneliness and cultural differences poses a future research challenge, considering the diversity of cultures.

The prevalence of social media use among young adults is high, and it has become a daily communication tool (Auxier and Anderson 2021; Sakurai et al. 2021). Previous studies indicate that lonely individuals frequently use the internet and email (Morahan-Martin and Schumacher 2003), and online communication has been reported to increase feelings of loneliness compared to face-to-face communication (Hu 2009). However, reports suggest no association between social media use and loneliness (Yavich et al. 2019). In a Japanese study examining the relationship between social media use and loneliness and traditional forms of communication, such as face-to-face and phone communication (Sakurai et al. 2021), frequent Twitter use was found to increase loneliness in young adults. However, the frequency of LINE utilization, a popular messaging application in Japan, was not related to loneliness. This suggests that the association between social media use and loneliness may vary depending on the type and usage patterns of social media.

Moreover, in the same study, infrequent traditional forms of communication, such as face-to-face and phone communication, were associated with higher loneliness in young adults (Sakurai et al. 2021). This finding highlights the significance of face-to-face and phone communication among young adults. Face-to-face communication is a primary way for individuals to interact. It involves non-verbal messages such as facial expressions and gestures, enabling rich communication. While nonverbal information is more limited than face-to-face communication, phone communication enables the reception of nonverbal messages through the tone of voice. Online communication lacks nonverbal messages, similar to face-to-face communication; however, it enables the creation of broad interpersonal networks.

From a temporal and spatial perspective, face-to-face and phone communication may involve burdens, such as coordinating schedules and finding a quiet place to talk. Conversely, online communication enables both parties to interact conveniently, making it a convenient communication method.

Loneliness is inherently tied to social connections, with interpersonal communication being a key factor. However, studies on loneliness and communication in young adults are notably more limited than those on older adults. Favotto's study with Canadian adolescents emphasizes the pivotal role of family communication in mitigating loneliness among young adults, highlighting its relevance to mental health and computer-mediated communication (CMC) (Favotto et al. 2019). Ying's research on Chinese immigrant young adults underscores the mitigating role of parent–child communication and parental warmth in reducing the adverse effects of economic pressures on young adults' loneliness (Ying et al. 2019). Additionally, Appel's study of Austrian secondary school students reveals a significant association between internet use and heightened loneliness, with parental support mediating this relationship (Appel et al. 2012). Overall, insights from these studies emphasize the indispensable role of family communication and parental support in understanding the influence of these factors on young adults' loneliness. Despite these insights, research exploring the relationship between loneliness and various communication modalities (such as face-to-face, telephone, and online communication) among young adults with individuals outside the family is currently lacking.

Addressing this deficiency is crucial for several reasons. First, understanding the nuanced interplay between loneliness and the frequency of face-to-face, telephonic, and online communication is essential in the context of societal shifts and technological advancements. Given the pervasive influence of recent social transformations, particularly the surge in online communication, meticulous exploration is needed to unravel how the diverse forms of communication contribute to the emergence of loneliness among individuals in their 20s. Second, exploring the association between loneliness and the frequency of communication modalities is crucial for devising targeted preventive and mitigation

measures. A comprehensive understanding of how face-to-face, online, and telephone interactions affect loneliness forms the foundation for crafting effective strategies to address this pressing issue.

Hence, this study aims to bridge this research gap by examining the relationship between face-to-face, telephone, and online communication frequencies and loneliness among young Japanese adults in their 20s, utilizing a comprehensive national survey dataset on loneliness and social isolation. Moreover, this study clarifies the relationship between high loneliness levels in young adults in their 20s and the frequency of use based on communication types, providing valuable insights that can inform interventions and policies aimed at reducing loneliness and fostering stronger social connections in this demographic group.

## 2. Materials and Methods

### 2.1. Data Source and Study Design

This study utilizes data derived from a nationwide survey conducted by the Japanese Cabinet Office (Cabinet Secretariat Japan 2021–2022). The survey targeted a random sample of 20,000 individuals aged 16 years and older across Japan. Specifically focusing on individuals aged 20–30 years, the study constitutes 1935 respondents within the dataset, all of whom participated, resulting in a robust effective response rate of 93.6% (1812 individuals). Notably, this research comprises a secondary analysis conducted following Japan's Statistical Laws, expressly permitted for purposes related to public interest and academic research.

### 2.2. Measurements

#### 2.2.1. Demographic Characteristics

Demographic attributes, including age, sex, living status, employment status, and health status, were evaluated. Sex was coded as "male = 1", and "female = 2". Living status was categorized into "living alone = 1", and "living with others = 2". Employment status was determined by classifying individuals engaging in any income-generating work or academic pursuits during the one-month survey period as "permanent or full-time = 1" and those not working or working part-time as "non-permanent or part-time = 2". Health status was assessed on a 5-point Likert scale, ranging from "excellent = 1", "good = 2", "fair = 3", "poor = 4", and "very poor = 5".

#### 2.2.2. Dependent Variable

Loneliness, the dependent variable, was measured using the UCLA Loneliness Scale (LS) version 3 (Arimoto and Tadaka 2019). Developed by Russell (Russell et al. 1978), the UCLA-LS is a globally employed scale for assessing loneliness as well as the unpleasant experience that occurs when an individual's network of social relations is deficient in some important way, either quantitatively or qualitatively (Peplau and Perlman 1982), with applications and validations in Australia (Elphinstone 2018), Northern Ireland (Shevlin et al. 2015), Italy (Boffo et al. 2012), and Japan (Masuda et al. 2012). While Russell's original scale comprised 20 and 10 items, maintaining high convergent validity and internal consistency, this study employed a 3-item, 4-point version specifically designed for written surveys, demonstrating reliability and validity equivalent to the original 20- and 10-item versions (Arimoto and Tadaka 2019). The three items of the UCLA-LS3 were: "How often do you feel that you lack companionship?", "How often do you feel left out?", and "How often do you feel isolated from others?" The response options were never, rarely, sometimes, and always, corresponding to 1, 2, 3, and 4, respectively, on the 4-point scale. The loneliness status of the total scores (3–12) was categorized as never: 3 points; almost never: 4 to 6 points; sometimes: 7 to 9 points; and always: 10–12 points.

2.2.3. Independent Variables

The independent variables included demographic characteristics and communication factors. Communication modalities were categorized as face-to-face, telephone, and online, with the latter encompassing social networking, email, and short messages. Communication frequency was assessed on a 7-point scale, categorized as "4–5 times a week = 1", "2–3 times a week = 2", "once a week = 3", "approximately once every 2 weeks = 4", "approximately once a month = 5", "less than once a month = 6", and "never = 7" (Cabinet Secretariat Japan 2021–2022).

*2.3. Statistical Analysis*

Descriptive statistics were used to characterize demographic features and communication modalities. Following the univariate analysis, to investigate the association between high levels of loneliness, communication styles, and demographic characteristics, we employed forced-entry multiple logistic regression analysis, calculated odds ratios (OR), and 95% confidence intervals (CI). Loneliness was treated as the dependent variable, whereas age, sex, living status, health status, employment status, and communication style were entered as independent variables. Adjustment for the confounding variables of demographic characteristics, including age, living situation, and health status, showed a significant association with loneliness.

*2.4. Ethical Approval*

The study was conducted in accordance with the 1964 Declaration of Helsinki (and its amendments) and was approved by the Institutional Review Board of the School of Health Sciences, Hokkaido University (Protocol No. 23–66; 2 October 2023). Additionally, it received approval from the Japan Cabinet Secretariat (Protocol No. 605; 25 August 2023) in compliance with Japanese Statistical Laws.

**3. Results**

*3.1. Demographic Characteristics and Communication of the Participants*

Table 1 provides the demographic characteristics and communication of the 1812 participants, comprising 811 males and 1001 females. The overall mean age of the participants was 24.8, SD = 2.9 years. Noteworthy demographics included 22.4% of single-person households. The health status distribution revealed that 29.7% reported fair health, 11.3% reported poor health, and 2.9% reported very poor health. Additionally, 27.0% of participants were engaged in non-permanent or part-time employment. Importantly, no statistically significant differences were observed between males and females regarding their demographic characteristics. Furthermore, concerning communication modalities, online interactions were the most prevalent, with 48.7% engaging 4–5 times per week or more. No significant gender differences were identified in communication patterns.

**Table 1.** Demographic characteristics and communication of the participants.

| | | Total, *n* = 1812 | | Men, *n* = 811 | | Women, *n* = 1001 | |
|---|---|---|---|---|---|---|---|
| | | **Mean** | **SD** | **Mean** | **SD** | **Mean** | **SD** |
| Age | | 24.8 | 2.9 | 24.8 | 3.0 | 24.8 | 2.9 |
| | | *n* | % | *n* | % | *n* | % |
| Living status | Living with others | 1393 | 76.9 | 612 | 75.5 | 781 | 78.0 |
| | Living alone | 405 | 22.4 | 192 | 23.7 | 213 | 21.3 |
| | Missing data | 14 | 0.8 | 7 | 0.9 | 7 | 0.7 |

**Table 1.** *Cont.*

| | | Total, *n* = 1812 | | Men, *n* = 811 | | Women, *n* = 1001 | |
|---|---|---|---|---|---|---|---|
| | | **Mean** | **SD** | **Mean** | **SD** | **Mean** | **SD** |
| Health status | Excellent | 552 | 30.5 | 258 | 31.8 | 294 | 29.4 |
| | Good | 462 | 25.5 | 180 | 22.2 | 282 | 28.2 |
| | Fair | 538 | 29.7 | 251 | 30.9 | 287 | 28.7 |
| | Poor | 204 | 11.3 | 95 | 11.7 | 109 | 10.9 |
| | Very poor | 52 | 2.9 | 25 | 3.1 | 27 | 2.7 |
| | Missing data | 4 | 0.2 | 2 | 0.2 | 2 | 0.2 |
| Employment status | Permanent or full-time | 1322 | 72.9 | 624 | 76.9 | 698 | 69.7 |
| | Non-permanent or part-time | 489 | 27.0 | 187 | 23.1 | 302 | 30.2 |
| | Missing data | 1 | 0.1 | – | – | 1 | 0.1 |
| Communication | | *n* | % | *n* | % | *n* | % |
| Face to face | 4–5 times a week or more | 340 | 18.8 | 152 | 18.7 | 188 | 18.8 |
| | Approximately 2–3 times a week | 251 | 13.9 | 106 | 13.1 | 145 | 14.5 |
| | Approximately once a week | 235 | 13.0 | 101 | 12.5 | 134 | 13.4 |
| | Approximately once every 2 weeks | 199 | 11.0 | 73 | 9.0 | 126 | 12.6 |
| | Approximately once a month | 272 | 15.0 | 118 | 14.5 | 154 | 15.4 |
| | Less than once a month | 278 | 15.3 | 117 | 14.4 | 161 | 16.1 |
| | Not at all | 237 | 13.1 | 144 | 17.8 | 93 | 9.3 |
| Phone call | 4–5 times a week or more | 197 | 10.9 | 90 | 11.1 | 107 | 10.7 |
| | Approximately 2–3 times a week | 238 | 13.1 | 123 | 15.2 | 115 | 11.5 |
| | Approximately once a week | 264 | 14.6 | 108 | 13.3 | 156 | 15.6 |
| | Approximately once every 2 weeks | 193 | 10.7 | 76 | 9.4 | 117 | 11.7 |
| | Approximately once a month | 274 | 15.1 | 107 | 13.2 | 167 | 16.7 |
| | Less than once a month | 278 | 15.3 | 122 | 15.0 | 156 | 15.6 |
| | Not at all | 368 | 20.3 | 185 | 22.8 | 183 | 18.3 |
| Online | 4–5 times a week or more | 883 | 48.7 | 332 | 40.9 | 551 | 55.0 |
| | Approximately 2–3 times a week | 320 | 17.7 | 133 | 16.4 | 187 | 18.7 |
| | Approximately once a week | 175 | 9.7 | 82 | 10.1 | 93 | 9.3 |
| | Approximately once every 2 weeks | 143 | 7.9 | 79 | 9.7 | 64 | 6.4 |
| | Approximately once a month | 122 | 6.7 | 78 | 9.6 | 44 | 4.4 |
| | Less than once a month | 80 | 4.4 | 47 | 5.8 | 33 | 3.3 |
| | Not at all | 89 | 4.9 | 60 | 7.4 | 29 | 2.9 |

### 3.2. Descriptive Analysis of Loneliness in the Participants

Table 2 shows the participants' loneliness. The UCLA-LS mean score for all participants was 6.7 (SD = 2.4). Furthermore, the distributions of reported loneliness levels, categorized as never, almost never, sometimes, and always, were 15.5%, 31.7%, 43.0%, and 9.7%, respectively. Remarkably, no statistically significant differences were found between men and women in the loneliness category.

**Table 2.** Descriptive analysis of loneliness in the participants.

| | Total, *n* = 1812 | Men, *n* = 811 | Women, *n* = 1001 |
|---|---|---|---|
| UCLA-LS (scores), mean ± SD | | | |
| | 6.7 ± 2.4 | 6.7 ± 2.4 | 6.7 ± 2.4 |
| UCLA-LS (category), *n* (%) | | | |
| Never | 281 (15.5) | 131 (16.2) | 150 (15.0) |
| Almost Never | 575 (31.7) | 242 (29.8) | 333 (33.3) |
| Sometimes | 780 (43.0) | 353 (43.5) | 427 (42.7) |
| Always | 176 (9.7) | 085 (10.5) | 91 (9.1) |

UCLA-LS (category): Never: 3 points; Almost never: 4 to 6 points; Sometimes: 7 to 9 points; and Always: 10–12 points.

### 3.3. Univariate Correlation between Loneliness and Factors in the Participants

Table 3 illustrates the univariate correlation between loneliness and various factors among the study participants. Across all participants, a significant increase in loneliness was observed with health status and non-regular employment, and a decrease in communication frequency across all modalities. Specifically, no significant association was identified between age and loneliness in women.

**Table 3.** Univariate correlation between loneliness and factors in the participants.

| | | Total, *n* = 1812 | | Men, *n* = 811 | | Women, *n* = 1001 | |
|---|---|---|---|---|---|---|---|
| | | Coefficient | *p* | Coefficient | *p* | Coefficient | *p* |
| Age | | 0.059 | 0.012 | 0.100 | 0.004 | 0.025 | 0.432 |
| Living status [a] | | −0.002 | 0.936 | −0.005 | 0.898 | 0.000 | 0.996 |
| Health status [b] | | 0.480 | 0.000 | 0.490 | 0.000 | 0.472 | 0.000 |
| Employment status [c] | | 0.131 | 0.000 | 0.146 | 0.000 | 0.120 | 0.000 |
| Communication | Face to face [d] | 0.261 | 0.000 | 0.257 | 0.000 | 0.266 | 0.000 |
| | Phone call [d] | 0.146 | 0.000 | 0.194 | 0.000 | 0.104 | 0.001 |
| | Online [d] | 0.242 | 0.000 | 0.242 | 0.000 | 0.251 | 0.000 |

Spearman's rank correlation coefficient. [a] No cohabitant = 1, Yes = 2; [b] Excellent = 1, Good = 2, Fair = 3, Poor = 4, Very Poor = 5; [c] Permanent = 1, Non-permanent = 2; [d] 4–5 times a week or more =1, approximately 2–3 times a week = 2, approximately once a week = 3, approximately once every 2 weeks = 4, approximately once a month = 5, less than once a month = 6, not at all = 7.

### 3.4. Impact of Communication on Loneliness in Participants

Table 4 presents the main results of the multiple logistic regression analysis examining the relationship between loneliness and communication style among young adults. Even after adjusting for demographic characteristics, higher loneliness was significantly associated with nonregular employment or part-time work (OR, 1.469; 95% CI: 1.190–1.813, *p* = 0.000) and poor health status (OR: 2.292, 95% CI: 2.070–2.539, *p* = 0.000). Poor health status more significantly impacted loneliness than non-permanent or part-time employment. Regarding loneliness and communication styles, a lower frequency of use was significantly associated with increased loneliness in all types, with online communication (OR: 1.469, 95% CI: 1.190–1.813, *p* = 0.000), face-to-face communication (OR: 1.469, 95% CI: 1.190–1.813, *p* = 0.000), and phone communication (OR: 1.469, 95% CI: 1.190–1.813, *p* = 0.000) having a sequential impact.

**Table 4.** Impact of communication on loneliness in participants: a multivariate logistic regression analysis.

| Independent Variables | *p* | OR | 95%CI |
|---|---|---|---|
| Age | 0.064 | 1.030 | 0.998–1.063 |
| Gender [a] | 0.338 | 0.913 | 0.759–1.100 |
| Living status [b] | 0.658 | 0.951 | 0.762–1.187 |
| Employment status [c] | 0.000 | 1.469 | 1.190–1.813 |
| Health status [d] | 0.000 | 2.292 | 2.070–2.539 |
| Online communication * [e] | 0.000 | 1.201 | 1.128–1.278 |
| Face-to-face communication * [e] | 0.000 | 1.155 | 1.096–1.216 |
| Phone call communication * [e] | 0.003 | 1.080 | 1.026–1.137 |

* Multivariate logistic analysis adjusting for age, gender, living status, employment status, and health status. Note: "1" for UCLA Loneliness Scale 7 to 12 points and "0" for UCLA Loneliness Scale 3 to 6 points. [a] Male = 1, Female = 2; [b] No cohabitant = 1, Yes = 2; [c] Permanent = 1, Non-permanent = 2; [d] Excellent = 1, Good = 2, Fair = 3, Poor = 4, Very Poor = 5; [e] 4–5 times a week or more = 1, approximately 2–3 times a week = 2, approximately once a week = 3, approximately once every 2 weeks = 4, approximately once a month = 5, less than once a month = 6, not at all = 7.

## 4. Discussion

### 4.1. Novelty and Strengths of This Study

This pioneering study, the first in Japan and the second in advanced countries following the United Kingdom, focuses on loneliness and isolation. Using a nationwide government-conducted survey, it investigates the association between face-to-face, phone, and online communication frequencies and heightened loneliness among individuals in their 20s. The findings reveal a significant correlation between loneliness and communication frequency, notably emphasizing the impact of online communication. Despite adjusting for demographic characteristics, a decline in communication frequency across all modalities correlates with increased loneliness. Among these modalities, online communication has the most substantial impact, followed by face-to-face communication, with phone communication exhibiting the least impact. This study presents a policy- and academic-driven investigation using an extensive dataset derived from the Cabinet Office of Japan's nationwide survey (Cabinet Secretariat Japan 2021–2022). As we navigated the complexities of loneliness, our study zeroes in on a randomly sampled cohort of 20,000 individuals aged 16 and older, offering a representative cross-section of the Japanese population. This method bolsters the strength of our findings and facilitates nuanced insights into the realms of loneliness and communication.

### 4.2. The Relationship between Loneliness and Demographic Characteristics

Loneliness among young Japanese adults in their 20s was significantly associated with age, sex, employment, and health status. However, findings on the relationship between loneliness and employment status are inconclusive. In a survey of individuals aged 19–34 in the Netherlands, 75.2% were employed, and no association between employment status and feelings of loneliness existed (Franssen et al. 2020). Conversely, a study in Germany that targeted individuals from youth to old age found that, among young adults aged 18–29, those in regular employment and the unemployed reported significantly higher levels of loneliness than those in non-regular employment (Luhmann and Hawkley 2016). This study also highlighted that the impact of employment on loneliness varies across age groups (Luhmann and Hawkley 2016). Notably, in a German study of young adults, the rate of regular employment was 38.3%, whereas in the current study, permanent or full-time employment was 72.9%. This indicates that the association between employment status and loneliness in young adults may vary owing to sociocultural norms regarding employment and job situations in different countries or regions.

Previous studies have consistently established the correlation between heightened feelings of loneliness and poor health. An investigation of Swiss youth aged 15–29 demonstrated a link between elevated levels of loneliness and unfavorable self-rated health (Richard et al. 2017). Additionally, a review commissioned by the UK government revealed that loneliness in young adults is associated with a decline in self-rated health, chronic headaches, stomach pain, sleep disturbances, and heightened morning fatigue (United Kingdom 2023). The impact of loneliness on human health is evident at the cellular (Cudjoe et al. 2022) and societal (Berkman et al. 2000) levels. This underscores the intricate relationship between loneliness and the developmental trajectory of health in young adults, emphasizing the imperative for a comprehensive understanding and development of robust preventive and supportive measures.

### 4.3. The Relationship between Loneliness and Communication Modalities

The heightened sense of loneliness among young adults in their 20s remained statistically significant even after adjusting for demographic characteristics, revealing a robust association with reduced frequencies of face-to-face, telephone, and online communication. This study underscores the critical role of all communication modalities, emphasizing their pivotal significance in preventing and alleviating loneliness among young adults in their 20s.

The correlation between loneliness in young adults and the frequency of social contact in interpersonal relationships aligns with meta-analytic findings, indicating that contact frequency is consistently linked to loneliness, irrespective of age (Luhmann and Hawkley 2016). Additionally, empirical evidence suggests that, as the frequency of contact with friends diminishes, loneliness intensifies among young adults (Franssen et al. 2020; Nyqvist et al. 2016). For the younger generation, maintaining a higher frequency of social relationships is indispensable for preventing loneliness (Victor and Yang 2012). Furthermore, a survey conducted in Japan during the pandemic revealed that individuals who engaged in communication with friends through calls or online interactions at least once a month reported lower levels of loneliness than those who did not (Arakawa et al. 2023). Conversely, some Japanese studies reported no significant association between loneliness in young adults aged 18–39 and the frequency of face-to-face or online social networking site (SNS) usage (Sakurai et al. 2021). While a stringent comparison between this study and Sakurai's research is challenging owing to differences in age groups, measures of loneliness, and communication styles, it implies that the relationship between loneliness and communication involves the frequency and quality, as well as depth, of communication. Meaningful and profound dialogue in online settings may reduce loneliness.

Regarding odds ratios for loneliness in young adults in their 20s based on communication type, online communication exhibited a slightly larger impact than face-to-face and telephone interactions. In this study, online communication emerged as the most prevalent, with 48.7% reporting its occurrence four to five times per week. The widespread adoption of online communication by young adults may explain its substantial impact on loneliness. The asynchronous nature of online communication, providing temporal and spatial flexibility without constraints on oneself and others, likely accounts for its pronounced effects. Additionally, online communication enables expressive elements such as emojis and stamps, offering enhanced possibilities for emotional expression compared with telephone and face-to-face interactions. The use of Social Networking Services (SNS) to address the loneliness of young individuals has been acknowledged to have various advantages and disadvantages (Sakurai et al. 2021). This highlights the importance of harmoniously integrating SNS-based communication with various forms of social interaction. Achieving a well-balanced integration of these approaches is crucial for alleviating loneliness in young individuals and promoting positive mental health outcomes. Notably, it is essential to appropriately combine different communication methods.

*4.4. Limitations*

This study has several limitations. First, owing to its cross-sectional design, establishing causal relationships between loneliness and demographic or communication-related factors proved unfeasible. Therefore, further investigations using longitudinal or interventional studies are imperative. Second, the reliance on participants' self-reports introduces potential subjective biases, with the possibility of overestimation or underestimation. Third, because the study focused exclusively on young adults in Japan, caution must be exercised when generalizing the findings to other countries or regions with different cultural backgrounds and social norms. Third, the rapidly evolving landscape of online communication, characterized by diverse types and uses, may have an evolving impact on loneliness among young adults.

**Author Contributions:** Conceptualization. E.T., Y.I., Y.T; methodology, N.O.; software, N.O.; validation., Y.T., Y.I., N.O., and E.T.; formal analysis, N.O.; investigation, E.T.; resources, E.T.; data curation, Y.T., Y.I., N.O., and E.T.; writing—original draft preparation, Y.T.; writing—review and editing, Y.T., Y.I., N.O., and E.T.; visualization, Y.I., and N.O., supervision, E.T.; project administration, E.T.; funding acquisition, E.T. All authors have read and agreed to the published version of the manuscript.

**Funding:** The sources of funding for our study were Grants-in-Aid for Scientific Research (KAKENHI) of the Japan Society for the Promotion of Science Grant Number 23H0321903 (PI: Etsuko TADAKA).

**Institutional Review Board Statement:** The study was conducted in accordance with the Declaration of Helsinki and was approved by the Institutional Review Board of the School of Health Sciences, Hokkaido University (Protocol No. 23-66; 28 September 2023). Additionally, it received approval from the Japan Cabinet Secretariat (Protocol No. 605; 25 August 2023) in compliance with the Japanese Statistical Laws.

**Informed Consent Statement:** All participants in this study provided written Informed Consent, affirming their voluntary participation rights or the right to decline, secured through the assurance of complete anonymity. Additionally, written Informed Consent was obtained for the disclosure of the survey results.

**Data Availability Statement:** The data utilized in this study emanate from the National Survey which is under the custodian-ship of the Cabinet Office of Japan. Access to the comprehensive dataset is contingent upon adherence to the procedures outlined in the Statistics Act of Japan. Prospective users are required to undergo an application process, seeking approval for data access, in accordance with the statutory mandates. Permission is granted solely under circumstances where the proposed utilization of the data aligns with the public interest, such as the generation of new statistics or contributions to academic research. For detailed information on the application procedures, please refer to the Statistics Bureau of Japan website at https://www.stat.go.jp/english/index.html, accessed on 12 December 2023.

**Conflicts of Interest:** The authors express their gratitude to all individuals who gave their time and energy to participate in this study.

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
