# Peer review of "Examining Loneliness: A Comparative Analysis of Face-to-Face, Telephone, and Online Communication among Japanese Young Adults"

_socsci, doi:10.3390/socsci13020076_

Round 1

Reviewer 1 Report

Comments and Suggestions for Authors

The theoretical introduction, although adequate, does not provide an in-depth review of previous studies conducted with young people on the relationships found between loneliness and all the variables included in the study. It would be useful to improve.

This article presents an analysis of secondary data from a large national survey conducted in Japan. The representativeness of the sample and the breadth of the sample are the strengths of this research. The methodology, sample and measures used are well described. The results achieved and their interpretation should be reviewed in depth.

It should be noted that the univariate correlations present a sign opposite to that expected theoretically and are interpreted in the opposite direction, given that the description of the measures informs that in all cases the higher the score the greater the quantity of the variable evaluated, these positive correlations are neither understandable nor interpretable. Please review in depth all the results achieved as well as their interpretation. 

Likewise, when the results of the Maike and Hawkley (2016) study are reported in the discussion, it is presented in a contradictory manner.  "A German  survey of individuals aged 18 to 29 found that feelings of loneliness were significantly higher among those in regular employment and the unemployed compared to those in non-regular employment (Maike and Hawkley 2016). These results suggest the potential for stable employment to have a protective effect on the loneliness experienced by young adults. The stability provided by regular employment may contribute to the development of social connections and economic stability, potentially alleviating feelings of loneliness." So, does regular employment reduce feelings of loneliness or not in Maike and Hawkley's study? Is this what their data indicate? The discussion of the results found should be improved after a review of previous studies and the statistics used. 

Author Response

Dear Reviewer 1

Thank you very much reviewing the manuscript. We appreciate your time and consideration. Please find attached response.

We include our point-by-point responses to each of the comments of the reviewers, as well as your comments.

Thank you for your continued consideration.

Reviewer 2 Report

Comments and Suggestions for Authors

The ms addresses an important topic, loneliness among young adults. The study presents a comparative study with an extensive sample. However, I have some concerns regarding the clarity of the concepts and methods. I hope my comments below will help the author(s) to continue their work.

First of all, I think that the some more specific definition of loneliness is needed, and how it can affect individuals psychologically and physically. There are an extensive body of research related to loneliness, for instance by Fardghassemi and Joffe (2021) and Becker et al. (2021), but there are many more.

In addition, when the context is Japanese young adults, mentioned also in title, I recommend adding some discussion about cultural differences in experiencing loneliness. Now the cultural aspect is missing, both in the introduction and in the discussion.

On page 2, please elaborate how the types of communication (face-to-face, telephone and online) differ from each other and why they might affect in a different way the feelings of loneliness.

Please, present exact research questions for the study.

On page 3, the demographic attributes measured are presented. Please, elaborate in the introduction why these attributes might matter in the feelings of loneliness. In addition, describe in detail how they are measured.

On page 3, line 104-105 the scale of health status variable is presented. Please, indicate which option corresponds the values 1, 2 etc. Now the reader does not understand does high score in health status indicate poor or good health.

On page 3, lines 123-124 the demographic attributes are repeated, I think unnecessarily.

Please, clarify also for the communication frequency variable does high score indicate high or low frequency.

Please, clarify what ± stands for age and for the loneliness score. I suppose it means standard deviation, but it could be explained.

In Table 3, it is difficult to interpret the correlations when the reader does not know it higher score means better or worse health and lower or higher communication frequency.

I recommend describing the use of logistic regression analysis more clearly. In addition, please, explain how to read the Table 4.

It would be helpful for the reader if the main findings of the study were presented clearly at the beginning of the discussion. Now it is somewhat difficult to figure out what the findings are and how the study contributes to the existing literature. Studies about different cultures (Germany and Netherlands) are mentioned in the discussion, but the issue is not discussed from the viewpoint of cultural differences.  I recommend extending the introduction as suggested above and discuss the results more deeply based on previous research.

References:

Becker, J.C., Hartwich, L. and Haslam, S.A. (2021), Neoliberalism can reduce well-being by promoting a sense of social disconnection, competition, and loneliness. Br J Soc Psychol, 60: 947-965. https://doi.org/10.1111/bjso.12438

Fardghassemi, S. & Joffe, H. (2021). Young adults experience of loneliness in London’s most deprived areas. Frontiers in Psychology, 12:660791, https://doi.org/10.3389/fpsyg.2021.660791

Author Response

Dear Reviewer2

Thank you very much for reviewing the manuscript. We appreciate your time and consideration. Please find attached a revised version.  

We include our point-by-point responses to each of the comments of the reviewers, as well as your comments.

Thank you for your continued consideration.

Yours sincerely,

Round 2

Reviewer 1 Report

Comments and Suggestions for Authors

The authors have responded adequately to suggestions and recommendations for improving the manuscript.

Author Response

Thank you very much for your comments. We are pleased that we were able to address your suggestions and improve the paper.

Reviewer 2 Report

Comments and Suggestions for Authors

Thank you for the revised manuscript. My comments and suggestions have been adequately considered. One minor point: it would be helpful for the reader if the scales of the variables were reported on measurements and not just as a footnote for Table 3. I mean the information for instance if excellent health represent value 1 or 5.

Author Response

Dear Reviewer2

Thank you very much for reviewing the manuscript entitled, “Examining Loneliness: A Comparative Analysis of Face-to-Face, Telephone, and Online Communication among Japanese Young Adults” (socsci-2800417). We appreciate your time and consideration.

We are grateful to know that it is potentially acceptable for publication in Social Sciences. Please find attached a revised version of our manuscript. The reviewers' comments helped us improve our manuscript to make it more understandable for our readers.

We look forward to hearing from you regarding our re-submission. We would be happy to respond to any further questions and comments that you may have.

Thank you for your continued consideration.
